# A Literature Review of Racial Disparities in Prostate Cancer Research

**Matthieu Vermeille [1], Kira-Lee Koster [2], David Benzaquen [3], Ambroise Champion [3], Daniel Taussky [3,4,*], Kevin Kaulanjan [5] and Martin Früh [2,6]**

1. Genolier Swiss Radio-Oncology Network, Clinique de Genolier, 1272 Genolier, Switzerland; vermeille@hin.ch
2. Department of Medical Oncology and Hematology, Cantonal Hospital St. Gallen, 9007 St. Gallen, Switzerland; kira-lee.koster@kssg.ch (K.-L.K.); martin.frueh@kssg.ch (M.F.)
3. Radiation Oncology, Hôpital de La Tour, 1217 Meyrin, Switzerland; david.benzaquen@latour.ch (D.B.); ambroise.champion@latour.ch (A.C.)
4. Department of Radiation Oncology, Centre Hospitalier de l'Université de Montréal, Montreal, QC H3Y2V4, Canada
5. Department of Urology, Université des Antilles, CHU de Guadeloupe, 97110 Pointe-à-Pitre, France; kevin.kaulanjan@gmail.com
6. Department of Medical Oncology, Inselspital, University Hospital Bern, 3010 Bern, Switzerland
* Correspondence: daniel.taussky.chum@ssss.gouv.qc.ca; Tel.: +514-890-8254; Fax: +514-412-7537

**Abstract:** Background: Despite recent awareness of institutional racism, there are still important racial disparities in prostate cancer medical research. We investigated the historical development of research on racial disparities and bias. Methods: PubMed was searched for the term 'prostate cancer race' and added key terms associated with racial disparity. As an indicator of scientific interest in the topic, we analyzed whether the number of publications increased linearly as an indicator of growing interest. The linearity is expressed as $R^2$. Results: The general search term "prostate cancer race" yielded 4507 publications. More specific search terms with $\geq 12$ publications showing a higher scientific interest were found after 2005. The terms with the most publications when added to the general term were "genetic" (n = 1011), "PSA" (n = 995), and "detection" (n = 861). There was a linear increase in publications for "prostate cancer race" ($R^2 = 0.75$) since 1980. Specific terms added to the general terms with a high linear increase ($R^2 \geq 0.7$) were "screening" ($R^2 = 0.82$), "detection" ($R^2 = 0.72$), "treatment access" ($R^2 = 0.71$), and "trial underrepresentation" ($R^2 = 0.71$). However, only a few studies have investigated its association with sexual activity. A combination with "sexual" showed 157 publications but only two years with $\geq 12$ publications/year. Conclusion: The terms "genetic", "PSA", and "detection" have been the focus of recent research on racial differences in prostate cancer. We found that old stereotypes are still being mentioned but seem to find little interest in the current literature. Further research interest was found in "treatment access". Recently, interest in socioeconomic factors has decreased.

**Keywords:** prostate cancer; racial disparity; detection; treatment access; socioeconomic factors

## 1. Introduction

There are large disparities in the incidence and mortality rates of prostate cancer among Black men [1,2]. It has been well-researched and the literature is vast. Black men are an average of two years younger than White men at diagnosis. Many studies have investigated the factors which might explain this racial difference. More recent publications concentrate not only on biological factors, such as genetics or environmental factors, but also on the structural and social determinants of health and equity, cultural mistrust, knowledge and communication, and other socioeconomic factors, such as insurance status and the interactions amongst different factors [2,3].

Scientific literature as far back as the 1930s has investigated the difference in the incidence of prostatic blockage between Black and White men. When reviewing the

literature, it seems that in the 1980s and the 1990s there were few studies investigating racial disparities, and the few articles investigating the subject were probably written mainly by White authors, focusing on biological differences in testosterone levels and sexual activity or sexually transmitted infections (STIs). This focus was based on the understanding that Black men have higher serum and intraprostatic tissue androgen concentrations [4–6].

Race and ethnicity are social constructs with limited utility in understanding medical research, but these terms may be useful for studying and viewing racism and disparities [7]. Historically, racial disparities have been attributed to biological differences. With time, there has been acknowledgment that racial differences in prostate cancer are rooted in many factors that are not only biological [8], but also starting with more subtle forms of racism and even going as far as institutional racism and systematic discrimination by the government, enterprises, schools, or other organizations.

First, we provide an overview of how the medical literature has evolved in recent years. Then, we investigate the factors over time that were of interest in explaining racial disparities in the medical community. Research on etiological factors for prostate cancer was analyzed and how it has changed over the years by analyzing differences in culture and sexual behavior in more scientific subjects, such as genetic and socioeconomic disparities.

In the second section, research papers on racial differences were cited, which in our opinion, are particularly biased. Racially biased research was frequent in the past but has continued to play a role until today.

## 2. Material and Methods

First, we reviewed the literature to find common key words associated with "prostate cancer" and "race". We then identified several historical articles in key publications that were among the first to report racial inequality in cancer detection and survival.

We accessed PubMed (https://pubmed.ncbi.nlm.nih.gov/) on 19 and 22 May 2023 and entered different search terms that have been in the past or are still attributed to racial disparities in prostate cancer.

We first searched the term "prostate cancer race", which yielded 4507 publications and then added additional more specific terms such as "genetic", "PSA", and "detection". Compared with the term "prostate cancer race", the search term "prostate cancer African descent" yielded 175 publications, and "prostate cancer ethnicity" yielded 3976 publications (Figure 1).

No limit was applied to the years analyzed. PubMed covers articles dating back to 1966 and selectively to 1865. Search terms were based on analyzing all 129 review articles on the terms "prostate cancer race" as accessed on 19 May 2023. While the number of citations an article generates would be a much better indicator of the interest that an article generates, we chose to use the number of publications on a certain topic as an indicator of scientific interest in a scientific subject. In our opinion, the number of publications equally represents interest in a subject. We chose to utilize PubMed and not other services such as for example the Web of Science, because a publication listed in PubMed means that the National Library of Medicine (NLM) deemed the scientific and editorial character and quality of a journal as meriting inclusion. We searched for each term, and recorded the number of publications. The year in which the term was first listed on PubMed was recorded, and then was analyzed at $\geq 6$ and $\geq 12$ publications. These numbers represent a certain interest in the scientific community in a subject indicated by there being at least one publication per month or per every other month in a year. We calculated the number of years between publication of $\geq 6$ and $\geq 12$ publications. A shorter time represents a rapid increase in interest in the subject's knowledge.

**Figure 1.** Flowchart of methods and materials.

Then, a graph based on the number of publications per year was created. The number of publications per year can easily be exported from PubMed to an Excel spreadsheet. Finally, Excel was used to calculate whether the increase in publications was linear. A linear increase represented an increase in interest and knowledge of the subject. Linearity is expressed as $R^2$, a measure of how well a linear regression model "fits" a dataset. The strong linearity was $R^2 > 0.7$, moderate 0.5–0.7, and weak was $R^2 = 0.3$–<0.5.

## 3. Results

First, we researched when awareness of racial differences must have begun. While reviewing the literature, we found that in general, the fact that there was an increasing incidence of prostate cancer in Black patients had been well-documented in large studies since at least 1980. Increasing incidence of prostate cancer in Black patients has been reported in several epidemiologic studies. An analysis of voluntary data supplied by hospitals in 1979 showed that Black patients presented with more advanced clinical stage and poorer survival rates. The first listed publication on "prostate cancer black mortality increase" appeared in PubMed in 1977. And the first time that there were >1 publications on the subject was in 1985. But only from 2008 on were there consistently >12 publications per year.

In our analysis of the literature in PubMed, we found among the 4507 publications on "prostate cancer race", most of the analyzed terms reached ≥6 publications/year beginning in 1994 (Table 1). Most search terms with ≥12 publications were identified after 2005. In recent years, publications with the terms "genetic" and "socioeconomic," for example, have more than 40 publications per year, showing that these subjects have been extensively researched.

**Table 1.** Factors researched on PubMed in combination with search terms added to "prostate cancer race" ordered according to number of publications.

| Term/Term Added | Publications May, 2023 | First Year Listed in Pubmed | Year First Time 6 Publicat. | Year First Time 12 Publicat. | Diff. in Years | $R^2$ |
|---|---|---|---|---|---|---|
| prostate cancer race | 4507 | 1964 | 1979 | 1990 | 11 | 0.75 |
| screening | 2246 | 1968 | 1983 | 1994 | 11 | 0.82 |
| genetic | 1011 | 1972 | 1995 | 1997 | 2 | 0.65 |
| PSA | 995 | 1991 | 1995 | 1995 | 0 | 0.48 |
| detection rate | 861 | 1980 | 1994 | 1997 | 3 | 0.72 |
| socioeconomic | 690 | 1971 | 2008 | 2013 | 5 | 0.80 |
| treatment outcome | 457 | 1993 | 1998 | 2000 | 2 | 0.43 |
| clinical trial | 355 | 1983 | 1996 | 2006 | 10 | 0.52 |
| treatment access | 311 | 1983 | 1996 | 2007 | 11 | 0.71 |
| active surveillance | 233 | 1996 | 2012 | 2012 | 0 | 0.57 |
| marital | 204 | 1972 | 1999 | 2013 | 14 | 0.58 |
| obesity | 191 | 1993 | 2005 | 2005 | 0 | 0.20 |
| diet | 185 | 1979 | 2003 | 2007 | 4 | 0.35 |
| belief | 177 | 1964 | 1998 | 2006 | 8 | 0.26 |
| testosterone | 112 | 1977 | 2004 | -- | -- | 0.09 |
| inflammation | 86 | 1996 | 2009 | -- | -- | 0.56 |
| immune | 77 | 2001 | 2014 | 2021 | 7 | 0.39 |
| microenvironnement | 26 | 2004 | 2021 | -- | -- | 0.25 |
| underrepresentation | 11 | 2001 | -- | -- | -- | 0.71 |
| microbiome | 9 | 2016 | -- | -- | -- | -- |
| stigma | 7 | 2015 | -- | -- | -- | -- |

Table 1 shows the number of publications per search term in descending order of the number of publications. The following terms: "PSA" (1995), "obesity" (2005), and "active surveillance" (2012), combined with "prostate cancer race" were picked up quickly by the scientific community, meaning that their number of publications immediately went from <6 to $\geq$12 per year.

In general, terms investigating an association with sexual activity, a term that could indicate a bias towards Black men, had few publications. Only two terms associated with sexual activity had >50 publications: the combination with "sexual" had 157 publications but only in two years (2011 and 2022) with exactly 12 publications, and "sexual factors" had 93 publications, but always <12 publications/year and a maximum of eight publications per year in 2013. The term "genetics testosterone" has been used in only 36 publications. This term is sometimes used to investigate genetic differences between the testosterone receptors. First publications about "testosterone", "sexual", and "marital" could be found in the 1970s (Table 2), while subjects that are still much discussed such as "treatment access", "clinical trial", and "treatment outcome" began only in the 1980s (Table 1).

Several search terms showed a strong linear increase ($R^2 > 0.7$) in publications, representing a continuous interest, such as the general search term "prostate cancer race" ($R^2 = 0.75$), "socioeconomic" (a factor that has recently been identified as a key factor for racial disparities in 690 publications), and a strong linear increase ($R^2 = 0.80$). Other specific terms added to the general term with a strong linear increase were "screening" ($R^2 = 0.82$), "detection" ($R^2 = 0.72$), "treatment access" ($R^2 = 0.71$), and "trial underrepresentation"

($R^2$ = 0.71). Interest in "genetics" has plateaued ($R^2$ = 0.65), but with more than 50 publications every year since 2015, after reaching ≥12 publications in 1997. Figure 2 lists some graphs illustrating the increase in publications of the four selected terms added to the general search term "prostate cancer race".

**Table 2.** Search terms added to "prostate cancer race" with sexual subjects that had <100 publications.

| Term Added | Publications May, 2023 | First Year Listed in Pubmed | Year First Time 6 Publicat. | Year First Time 12 Publicat. | Diff in Years | $R^2$ |
|---|---|---|---|---|---|---|
| sexual | 157 | 1971 | 2005 | 2011 | 6 | 0.36 |
| sexual factors | 93 | 1971 | 2005 | -- | -- | 0.33 |
| sexual activity | 45 | 1973 | 2011 | -- | -- | 0.14 |
| intercourse | 16 | 1974 | -- | -- | -- | 0.007 |
| STD | 4 | 1992 | -- | -- | | |

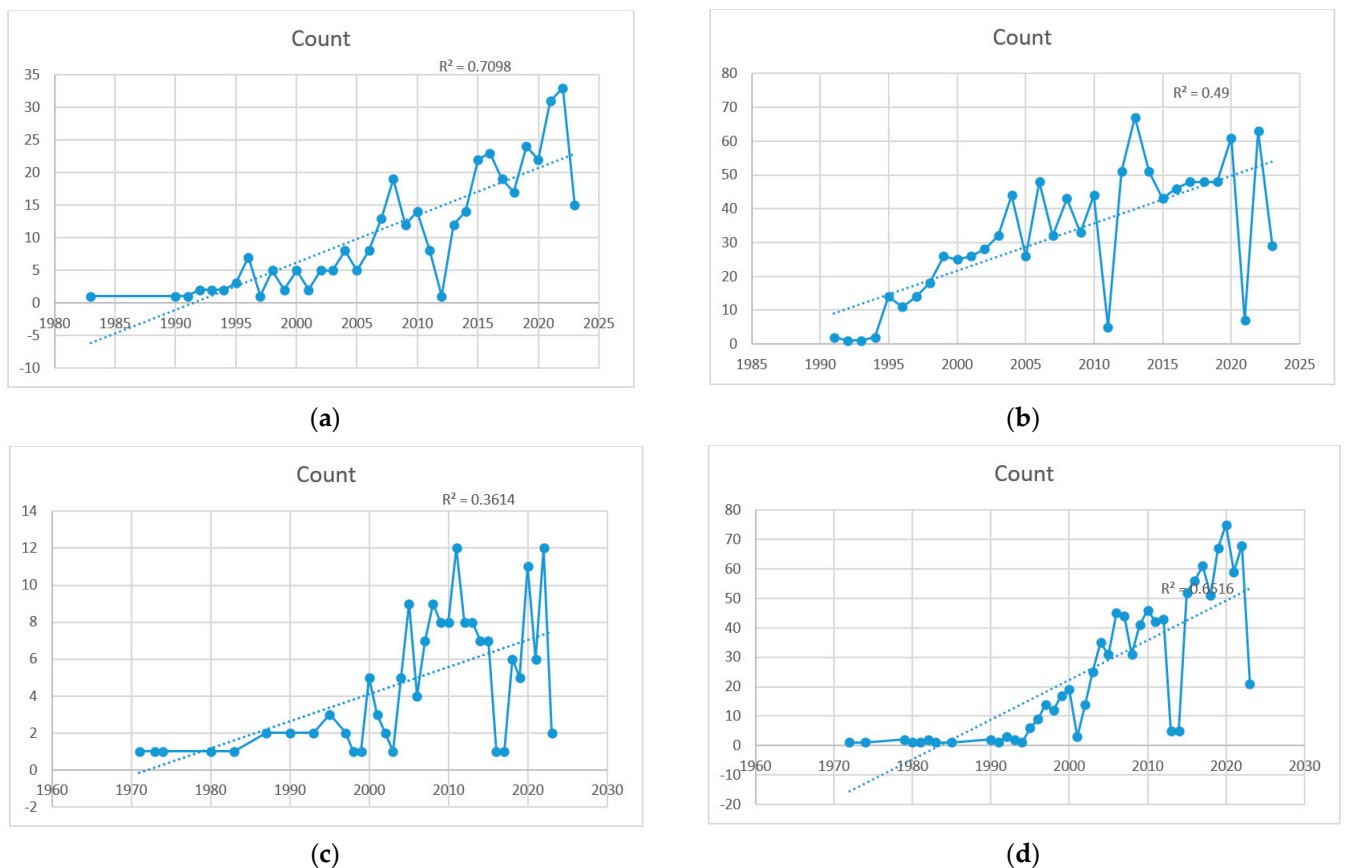

**Figure 2.** Selected terms added to the general search term "prostate cancer race". (**a**) "Treatment access"; (**b**) "PSA"; (**c**) "Sexual"; (**d**) "genetic".

## 4. Discussion

We found that Black men had more advanced disease and worse survival since at least 1980 when larger studies were published. But the subject of increased mortality only really gained interest in the scientific community from 2008 on, when there were consistently >12 publications per year on the topic.

Today, we found that there is a strong awareness of racial disparities in prostate cancer with >4500 publications in PubMed on the subject of "prostate cancer race". Interest in racial disparities seems to have begun around 1954, and most of the search terms with

≥12 publications were found after 2005. In the last years, publications with the terms "genetic" and "socioeconomic", for example, have had more than 40 publications/year, thereby implying that these subjects are thoroughly researched. Most articles deal with genetics, as well as terms associated with prostate cancer detection such as "PSA" and "detection" of prostate cancer. Recently, there has been a large and linear increase in the subject of "socioeconomic" factors and prostate cancer detection as well as "treatment access" and "trial underrepresentation". Factors traditionally associated with bias against Black men, such as sexual activity and testosterone levels, were of little but persistent interest.

### 4.1. Race in Prostate Cancer Research

Race has been recognized as a social construct defined by how one group perceives another [9]. However, a more recent publication argued that race has both genetic and social components. The authors embraced genetic studies in African populations to better understand these diseases [8,10]. In their 1992 review, Burks and Littleton, from the Henry Ford Hospital in Detroit [11], who focused on the epidemiology of prostate cancer in Black men, had already shown that significant racial differences are related to factors such as access to medical care, genetic and environmental factors, and cultural differences, including diet and social habits. They stated that most reports presented conflicting data with no clear positive correlations, and their conclusions are often speculative. Better controlled prospective studies of epidemiologic variables and a comprehensive genetic evaluation of Black families with prostate cancer are needed, to more thoroughly understand the racial disparity affecting American Black men and the biology of this disease in all men.

### 4.2. Comparison to Cervical Cancer

Compared to prostate cancer, cervical cancer is clearly associated with STIs such as ***Human papillomavirus*** (**HPV**) [12]. Interestingly, the search term "cervical cancer race" yielded 1990 publications compared with the 4500 terms for prostate cancer. Adding the search term "sexual" to cervical cancer resulted in 225 publications, compared with 157 publications for prostate cancer.

### 4.3. A Short History of the Importance of Race in Prostate Cancer Research

We examined the historical development of research on racial disparity. The opinion that hormonal and sexual factors play a role in the differences between Black and White men probably originated in the fact that since the early 1940s it has been known that prostate cancer is hormonally dependent [13]. Additionally, STIs had been investigated as an etiological factor for prostate cancer. One study found that circumcision appeared to be protective only among Black men [14] and another study found that HSV-2 might be implicated in prostate cancer development [15,16]. The theory that the presence of sexually transmitted infectious agents cause prostate cancer and thereby a history of venereal disease posing as a risk factor for prostate cancer, suggesting that racial disparity might be due to differences in sexual activity between Black and White men [6] has been abandoned [17].

Another factor contributing to the bias in the medical establishment regarding Black men is that Black patients are sometimes seen as more tolerant to pain in general [18] and urinary symptoms in particular, and therefore are significantly less likely than White patients to receive prescriptions for opioids [19]. A telling publication from South Africa published in 1966 reported that nearly two-thirds of African and half of Asiatic patients delayed coming to the hospital until forced to do so by the retention of urine. The reason given by the authors was the greater tolerance to dysuria [13].

Even back then, the much more obvious cause of delayed doctor visits would have likely been restricted access to the medical system or the perception of unequal treatment by Black patients, resulting in distrust in the system.

### 4.4. Bias against Black Patients in Prostate Cancer Research

A factor that has gained little interest today is that Black men seem, despite similar care settings, to be generally less satisfied with their treatment [20]. In their 2008 publication, one of >700 publications, Sanda et al. studied the quality of life of patients with prostate cancer. One of their findings was that "Black patients reported lower satisfaction with the degree of overall treatment outcomes". We were unable to identify the reasons for this discrepancy between Black and non-Black patients. However, this finding has received little research attention. The authors were unable to determine whether these differences were biological, a reflection of disparities in the quality of care, or differences in patient expectations. One could hypothesize that within the trial setting, patients may have felt uncomfortable, and their expectations and interaction with the research staff were not optimal, or there were other cultural or socioeconomic differences. This is one of the reasons why the Black population in clinical trials in general is underrepresented [21].

One could argue that the fact that they were treated in a trial could have created mistrust in patients, that their interaction with the research staff was not optimal, or that there were other cultural or socioeconomic differences in Black patients [22].

Some of the biased literature may stem from previous research on benign prostatic hyperplasia (BPH) in Black men. Derbes et al. in their findings published in 1937 in the *Journal of Urology* [23] showed that some of the biased literature may stem from previous research on BPH in Black men, which examined 1405 cases of BPH treated at the State of Louisiana Charity Hospital. They found that Black men possibly have a higher frequency of BPH and are, on average, five years younger than White men. They attributed these findings to the fact that it is generally supposed that the Black man "accepts physical discomfort, especially that associated with the genito-urinary tract, as a natural event in the course of life and would be slower in seeking medical aid than the White man". They cited other reasons that are not reproduced here.

There were studies without any racial prejudice such as the publication by Burns, read before the Southeastern Section, American Urological Association, in Biloxi, Miss, in 1939 [24].

Ross et al. published a rather strongly biased example in May 1987 in the *Journal of the National Cancer Institute* [6]. They stated that they investigated the reason for the higher risk of prostate cancer in Black men and stated that a history of venereal disease, as well as the frequency of sexual intercourse, was higher in Black men. Later on, they explained that these differences were because the two groups were dissimilar in "social class characteristics".

In 2022, Basourakos et al. [25] found that the age-specific frequencies of definitive treatment were similar for men of all races. Additionally, they found that "Black men have a higher incidence of and mortality from prostate cancer compared to men of other races, and that the Black race does not appear to be associated with inferior long-term outcomes as long as there is equal access to care and standardized treatment". There is increasing awareness of the necessity to include Black men in prostate cancer trials in general and specifically in cancer prevention trials [26].

### 4.5. Limitations of Our Study

The limitation of our study is that we did not separate publications about Black men only but included the term "race" without specifying further. It would have been a herculean task to analyze 4500 abstracts and classify each publication according to one main topic. Therefore, some publications have appeared in several searches although they have no direct importance to the subject of the publication. Furthermore, we did not include the search term "ethnicity" with nearly 4000 publications. We have, therefore, included some patients of origins other than African, although most studies on race and prostate cancer deal with disparities between Black men and other races. In general, little research has been conducted on other racial disparities outside North America. Therefore, our data are mostly US-specific and cannot necessarily be applied to other countries. There is a growing

number of journals, and therefore, publications and research, over the years. Therefore, more recent topics have more publications and a faster uptake per definition than older topics. Healthcare professionals have been shown to exhibit the same level of implicit bias as that of the wider population. It has been observed that patients who experience racism lack trust and experience a delay in seeking healthcare [27].

## 5. Conclusions

In conclusion, interest in the influence of race on prostate cancer began in the mid-1990s and has become a more researched subject since 2005. Biased terms dealing with racial sex differences have gained little but persistent interest. Recently, terms such as "genetic", "PSA", and "detection" have become the focus of research.

**Author Contributions:** Conceptualization: M.V., D.T., K.-L.K., and M.F.; Methodology: M.V., D.T., K.-L.K., and M.F.; Software: D.T.; Validation: D.T.; Formal Analysis: D.T.; Investigation: M.V., D.T., K.-L.K., and M.F.; Resources: D.T., A.C., and D.B.; Data Curation: D.T.; Writing—Original Draft Preparation: M.V., D.T., K.-L.K., M.F., A.C., D.B., and K.K.; Writing—Review and Editing: M.V., D.T., K.-L.K., M.F., A.C., D.B., and K.K.; Visualization: M.V., D.T., K.-L.K., M.F., A.C., D.B., and K.K.; Supervision: D.T.; Project Administration: D.T.; All authors have read and agreed to the published version of the manuscript.

**Funding:** This research received no external funding.

**Conflicts of Interest:** The authors declare no conflict of interest.

## Abbreviations

STIs: sexually transmitted infections; HPV: Human papillomavirus; BPH: benign prostatic hyperplasia.

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
