# Peer review of "A Literature Review of Racial Disparities in Prostate Cancer Research"

_curroncol, doi:10.3390/curroncol30110718_

Round 1

Reviewer 1 Report

Comments and Suggestions for Authors

This study was reported the racial trends in prostate cancer studies. Generally, this paper is well written. However, this study only shows the transition of papers on prostate cancer that have been published so far, and it does not have any depth of content. Therefore, it is unfortunately difficult for the reader to understand what we are trying to say through this paper.

Minor

1.    On line 50, the authors should use STI instead of STD.

Author Response

Reviewer 1

However, this study only shows the transition of papers on prostate cancer that have been published so far, and it does not have any depth of content. Therefore, it is unfortunately difficult for the reader to understand what we are trying to say through this paper.

We are sorry that this reviewer sees our paper so negatively. We believe that our manuscript is truly novel and has many interesting findings that can serve as a point of reference for future articles.

 Minor

  1. On line 50, the authors should use STI instead of STD.

We corrected this point

Reviewer 2 Report

Comments and Suggestions for Authors

Prostate cancer disparities represent an intensely debated subject of great importance which can guide the urologist's daily practice for cancer screening and follow-up.

There are some aspects that should be improved in the present manuscript.

1. Pay attention to the text. There are plenty of mistakes regarding punctuation, language, etc...

2. Try to change the "We" words from the phrases and introduce some more professional ones like "This study", and "The present manuscript" highlighted etc..

3. Please add a flow chart in the material and methods section.

4. Excel was the only analyzing program you used?

5. You can structure the discussion section into small subchapters to be more easy to follow.

6. Add an abbreviations list.

Comments on the Quality of English Language

Major text issues

Author Response

Reviewer 2

  1. Pay attention to the text. There are plenty of mistakes regarding punctuation, language, etc...

We had a native English-speaker correct our text

  1. Try to change the "We" words from the phrases and introduce some more professional ones like "This study", and "The present manuscript" highlighted etc..

We were able to replace many “we” in the text by other expressions

  1. Please add a flow chart in the material and methods section.

We added a flowchart in the material and methods section

  1. Excel was the only analyzing program you used?

Yes. Excel can very easily calculate if the increase in publications is linear or not. This is the only statistical analysis that we were using for this analysis. 

  1. You can structure the discussion section into small subchapters to be more easy to follow.

We added several subtitles in the discussion to make the discussion more comprehensible.

  1. Add an abbreviations list.

We added a list with abbreviations at the end of the manuscript.

Reviewer 3 Report

Comments and Suggestions for Authors

This manuscript aims to conduct an overview of how literature evolved regarding racial disparities of prostate cancer in recent years. However, the method to address this topic is over-simplistic and could be improved from the following aspects. 

1. The title of this manuscript is too general and seems too big. I would recommend including "literature review" or "medical literature comparison" in the title. 

2. Why use 6 and 12 as the two threshold cutoffs for the comparison? Please elaborate the reasons underlying this because the cutoffs would largely impact the results. 

3. Please provide more rationale of why the number of publications would be an indicator of the interest of the topic. There are other confounding factors such as the number of journals included in the database, or the increase in the number of researchers, etc. 

4. In the result section, when describing the keyword searching process, did the authors include all publications with the appearance of these keywords or the research described in the publication was directly related to those keywords? For example, in a prostate cancer genomic analysis publication, even though PSA is mentioned in the text maybe in the clinical data table, it does not make the publication one focusing on prostate cancer PSA. If the authors want to conduct a thorough analysis of different topics, please consider using a more evolved topic extraction model instead of searching by term. 

5. For the linear regression analysis, please include the statistical significane. 

6. For a more thorough analysis/comparison of how the interest of prostate cancer racial disparities, the authors should also highlight some of the key achievements in the chronicle mapping. For example, when was the first publication indicating that people of African ancestry had a higher mortality and diagnostic rate? Did more publications appear addressing racial disparities after that? 

Author Response

Reviewer 3

the method to address this topic is over-simplistic and could be improved from the following aspects. 

  1. The title of this manuscript is too general and seems too big. I would recommend including "literature review" or "medical literature comparison" in the title. 

Thank you for this suggestion. We changed the ttile to “Literature review of racial disparities research in prostate”

  1. Why use 6 and 12 as the two threshold cutoffs for the comparison? Please elaborate the reasons underlying this because the cutoffs would largely impact the results. 

We explained that These numbers represent a certain interest in the scientific community in a subject because there was at least one publication per month or per every other month in a year.  “

  1. Please provide more rationale of why the number of publications would be an indicator of the interest of the topic. There are other confounding factors such as the number of journals included in the database, or the increase in the number of researchers, etc

While we agree that the number of citations would be a much better indicator of the interest that an article generates, we chose to use the number of publications on a certain topic as an indicator of scientific interest in a scientific subject. In our opinion, the number of publications equally represents in interest in a subject. We chose to utilize pubmed and not other services such as for example the Web of Science, because a Publication listed in pubmed means that the National Library of Medicine (NLM) judged that the scientific and editorial character and quality of a journal merit its inclusion

  1. In the result section, when describing the keyword searching process, did the authors include all publications with the appearance of these keywords or the research described in the publication was directly related to those keywords? For example, in a prostate cancer genomic analysis publication, even though PSA is mentioned in the text maybe in the clinical data table, it does not make the publication one focusing on prostate cancer PSA. If the authors want to conduct a thorough analysis of different topics, please consider using a more evolved topic extraction model instead of searching by term. 

I It would have been a herculean task to analyze 4500 abstracts and classify each publication according to one main topic. Therefore some publications have appeared in several searches although they have no direct importance to the subject of the publication.

  1. For the linear regression analysis, please include the statistical significane. 

You can use a linear regression analysis to predict a certain variable or to “model the relationship between a scalar response and one or more explanatory variables”. We analyze whether there was a linear increase in the number of publications or not. This is not expressed as a “p” but as a R2, a measure of how well a linear regression model "fits” a dataset.  As stated in the methods section, a strong linearity is defined as a R2 >0.7, moderate 0.5 - 0.7, and weak as R2= 0.3 - <0.5.

  1. For a more thorough analysis/comparison of how the interest of prostate cancer racial disparities, the authors should also highlight some of the key achievements in the chronicle mapping. For example, when was the first publication indicating that people of African ancestry had a higher mortality and diagnostic rate? Did more publications appear addressing racial disparities after that

This is an excellent idea. We added in the methods section that: “First, we reviewed the literature to find common key words associated with “prostate cancer” and “race”. We then identified several historical articles in key publications that were among the first to report racial inequality in cancer detection and survival”

We added a paragraph at the beginning of the results section stating that: “First we researched when awareness of racial differences must have begun. While reviewing the literature, we found that in general, the fact that there was an increasing incidence of prostate cancer in Black patients, had been well documented in large studies since at least 1980. Increasing incidence of prostate cancer in black patients has been reported in several epidemiologic studies. An analysis of voluntary data supplied by hospitals in 1979 showed that Black patients presented with more advanced clinical stage and poorer survival rates. The first listed publication on “prostate cancer black mortality increase” appeared in pubmed in1977. And the first time that there were >1 publications on the subject was in 1985. But only from 2008 on were there consistently >12 publications per year.“

We also added a smaller paragraph at the beginning of the Discussion section saying that “We found that black men had more advanced disease and worse survival since at least 1980 when larger studies were published. But the subject of increased mortality only really gained interest in the scientific community from 2008 on, when there were consistently >12 publications per year on the topic.

Round 2

Reviewer 1 Report

Comments and Suggestions for Authors

none.

Reviewer 2 Report

Comments and Suggestions for Authors

All suggestions were accomplished.

I proposed the manuscript for publication.

Comments on the Quality of English Language

The English is fine!

Reviewer 3 Report

Comments and Suggestions for Authors

The authors have adequately addressed all previous comments. The added paragraph in the result section and the revised title would further emphasize the key findings of this study.